# Enhanced UV-B Radiation in Potato Stems and Leaves Promotes the Accumulation of Anthocyanins in Tubers

Lingyan Cui [1,2,3,†], Maoxing Li [1,2,3,†], Xing Zhang [1,2,3], Zongming Guo [3], Kaifeng Li [1,2,3], Yuhan Shi [1], Qiong Wang [3,*] and Huachun Guo [1,2,3,*]

1 College of Agronomy and Biotechnology, Yunnan Agricultural University, Kunming 650201, China; cuily20012005@163.com (L.C.); limaoxing2011@163.com (M.L.); bryan960614@163.com (X.Z.); dtllx04@163.com (K.L.); syhslack@163.com (Y.S.)
2 Yunnan Engineering Research Center of Tuber and Root Crop Bio-Breeding and Healthy Seed Propagation, Yunnan Agricultural University, Kunming 650201, China
3 Tuber and Root Crops Research Institute, Yunnan Agricultural University, Kunming 650201, China; gaze122@163.com
* Correspondence: 2000005@ynau.edu.cn (Q.W.); 1984013@ynau.edu.cn (H.G.)
† These authors contributed equally to this work.

**Abstract:** Enhanced ultraviolet-B (UV-B) radiation promotes anthocyanin biosynthesis in leaves, flowers and fruits of plants. However, the effects and underlying mechanisms of enhanced UV-B radiation on the accumulation of anthocyanins in the tubers of potatoes (*Solanum tuberosum* L.) remain unclear. Herein, reciprocal grafting experiments were first conducted using colored and uncolored potatoes, demonstrating that the anthocyanins in potato tubers were synthesized in situ, and not transported from the leaves to the tubers. Furthermore, the enhanced UV-B radiation ($2.5 \ kJ \cdot m^{-2} \cdot d^{-1}$) on potato stems and leaves significantly increased the contents of total anthocyanin and monomeric pelargonidin and peonidin in the red-fleshed potato '21-1' tubers, compared to the untreated control. A comparative transcriptomic analysis showed that there were 2139 differentially expressed genes (DEGs) under UV-B treatment in comparison to the control, including 1724 up-regulated and 415 down-regulated genes. The anthocyanin-related enzymatic genes in the tubers such as *PAL*, *C4H*, *4CL*, *CHS*, *CHI*, *F3H*, *F3'5'H*, *ANS*, *UFGTs*, and *GSTs* were up-regulated under UV-B treatment, except for a down-regulated *F3'H*. A known anthocyanin-related transcription factor *StbHLH1* also showed a significantly higher expression level under UV-B treatment. Moreover, six differentially expressed MYB transcription factors were remarkably correlated to almost all anthocyanin-related enzymatic genes. Additionally, a DEGs enrichment analysis suggested that jasmonic acid might be a potential UV-B signaling molecule involved in the UV-B-induced tuber biosynthesis of anthocyanin. These results indicated that enhanced UV-B radiation in potato stems and leaves induced anthocyanin accumulation in the tubers by regulating the enzymatic genes and transcription factors involved in anthocyanin biosynthesis. This study provides novel insights into the mechanisms of enhanced UV-B radiation that regulate the anthocyanin biosynthesis in potato tubers.

**Keywords:** UV-B radiation; potato; anthocyanin; in situ synthesis; transcriptome

## 1. Introduction

Potato (*Solanum tuberosum* L.) is an important staple and vegetable crop, known for its edible underground tubers. The skin and flesh color of the potato tuber is commonly white or yellow, while potatoes with red, purple, or blue skin and/or flesh are called colored potatoes, which represent an essential genetic resource for potato breeding [1,2]. In addition to nutrients, such as high starch, low fat, high vitamin C, and high dietary fiber, colored potatoes are rich in anthocyanins [2], granting them an antioxidant activity that is 2–3 times higher than uncolored potatoes [3]. Numerous researchers have documented that anthocyanins possess antioxidant attributes that enable them to effectively eliminate free radicals, thereby aiding in the safeguarding of individuals against the onset of aging,

diabetes, cardiovascular diseases, and cancers [4–6]. Anthocyanins are also a primary source of natural pigments, serving as beneficial substitutes for artificial dyes in food processing. Colored potato tubers with high levels of anthocyanin are a recognized health food, and are popular with consumers and researchers.

Anthocyanins, which belong to the flavonoid metabolites, are widespread in different tissues of plants, imparting on them a range of colors including red, purple, and blue [7,8]. The anthocyanin biosynthetic pathway has been well-studied in model plants and field crops [9,10], where it is synthesized through the phenylpropanoid and flavonoid metabolic pathways. Phenylalanine serves as a direct precursor for the biosynthesis of anthocyanins and other flavonoids, which are synthesized in the endoplasmic reticulum through a series of enzymatic modifications facilitated by enzymes such as phenylalanine ammonia-lyase (PAL), trans-cinnamate 4-monooxygenase (C4H), 4-coumarate-CoA ligase (4CL), chalcone synthase (CHS), chalcone isomerase (CHI), flavanone 3-hydroxylase (F3H), flavonoid 3'-hydroxylase (F3'H), flavonoid 3',5'-hydroxylase (F3'5'H), dihydroflavonol 4-reductase (DFR), anthocyanidin synthase (ANS), UDP-glucose:flavonoid 3-O-glucosyltransferase (UFGT), O-methyltransferase (OMT), and anthocyanin acyltransferase (AAT) [11]. After synthesis, anthocyanins are efficiently transported to the vacuoles, assisted by glutathione S-transferase (GST) or multidrug and toxic compound extrusion (MATE) [12]. Anthocyanin biosynthesis is regulated by the MBW protein complex composed of MYB, bHLH, and WD40, with MYB transcription factors playing a central role [13,14].

Anthocyanins protect plants against biotic and abiotic stress, including wounding, pest infection, low temperatures, drought, and ultraviolet-B (UV-B) radiation [15,16], playing a vital role in plant–environment interactions. UV-B radiation (280–315 nm) constitutes a portion of sunlight and exerts a significant influence on plant growth and development as both an abiotic stress and an environmental signal factor [16,17]. As an abiotic stress, UV-B usually induces thymine dimers in DNA, and breaks proteins, lipids, and the photosynthetic system, ultimately changing plant growth. Plants have evolved various effective defense mechanisms to prevent or minimize UV-B-induced damages, with the accumulation of flavonoids being one of these adaptation mechanisms [18]. Flavonoids, particularly anthocyanins, are involved in absorbing UV-B and scavenging the reactive oxygen species generated by enhanced UV-B radiation [16]. On the other hand, as an environmental signal, UV-B also participates in plant photomorphogenesis, such as inhibiting hypocotyl elongation, cotyledon expansion, and flavonoids accumulation [18]. Numerous studies have proven that UV-B induces the expression of genes involved in the flavonoid biosynthetic pathway, thus increasing the anthocyanin content of various plants, such as apples [19], blueberries [20], and grapes [21], as well as carrot taproots [22]. Moreover, the effects of UV-B on the biosynthesis and accumulation of plant flavonoids vary according to the dose and duration of exposure. Proper-dose UV-B radiation (5 kJ·m$^{-2}$) did not change the agronomic characters of potato plants significantly, but significantly increased the expression of structural genes regulating anthocyanins synthesis. A higher dose of UV-B radiation (10 kJ·m$^{-2}$) will cause greater damage to the colored potato plants, reducing the tubers' yield and nutrients [23]. Utilizing enhanced UV-B radiation to regulate biosynthesis and the accumulation of beneficial secondary metabolites, such as anthocyanins, in plants has become a point of interest.

In our previous research, we treated colored potatoes with different doses of UV-B radiation and found that a low dose of enhanced UV-B radiation (2.5 kJ·m$^{-2}$·d$^{-1}$) does not damage the growth of potato plants, and enhanced UV-B radiation, indeed, increases the activity of antioxidant enzymes and the concentration of secondary metabolites such as anthocyanins in the leaves of potato plants [24]. Recently, Liu et al. [14] reported that altitude significantly affected the flavonoid content in colored potato tubers, and that the anthocyanin content in colored potato tubers grown at high altitudes (3600 m) was significantly higher compared to in those grown at lower (800 m) and medium (1800 m) altitudes. Given the relationship between altitude and UV-B intensity, UV-B radiation could be related to the anthocyanin accumulation in the underground tubers of colored potatoes [23,24], although there is also a relationship between altitude and temperature,

and low temperatures also induce anthocyanin biosynthesis [25,26]. However, it is unclear whether the anthocyanin biosynthesis in the aboveground part (stems and leaves) of the potato is related to the anthocyanin in the underground tubers. The effects and potential mechanisms of enhanced UV-B exposure in potato stems and leaves on anthocyanin's accumulation in underground tubers are also not clear.

In the present study, we first conducted three independent reciprocal grafting experiments in groups of colored and uncolored potatoes. The results demonstrated that the anthocyanins in the underground tubers of potato were synthesized in situ rather than synthesized in the stems and leaves of the aboveground part and then transported to the underground tubers. We also studied the differences in anthocyanin accumulation in colored potato tubers grown under enhanced UV-B radiation and natural sunlight, and identified the DEGs related to UV-B-induced anthocyanin biosynthesis in the tubers through comparative transcriptome analysis and qRT-PCR analysis. This study lays the theoretical foundation for unraveling the molecular mechanism of enhanced UV-B radiation promoting the biosynthesis and accumulation of anthocyanins in potato tubers.

## 2. Materials and Methods

### 2.1. Plant Materials and Growth Conditions

The virus-free potato clones tested were provided by the Tuber and Root Crops Research Institute of Yunnan Agricultural University (Kunming, Yunnan, China). Potato clones '21-1' (red skin and red flesh), '21-3' (pale yellow skin and white flesh), '1417-5' (purple skin and purple flesh), and '15D1' (white skin and white flesh) are high-generation potato strains. 'Heijingang' (HJG, purple skin and purple flesh) and 'Lishu No.6' (LS6, white skin and white flesh) are selective breeding potato varieties in China. These six potato clones were used for grafting experiments (Figures 1 and S1), while clone '21-1' was also used in the UV-B radiation treatment. The experiment was carried out on the experimental farm (N 25°13', E 102°74') of Yunnan Agricultural University using open-air potted cultivation methods. The experimental farm belongs to a subtropical plateau-mountain monsoon climate zone, with an altitude of 1943 m. The average annual sunshine duration is 11.95 h, the average annual temperature is 14.7 °C, the average annual precipitation is 1000 mm, and the frost-free period is 301 d.

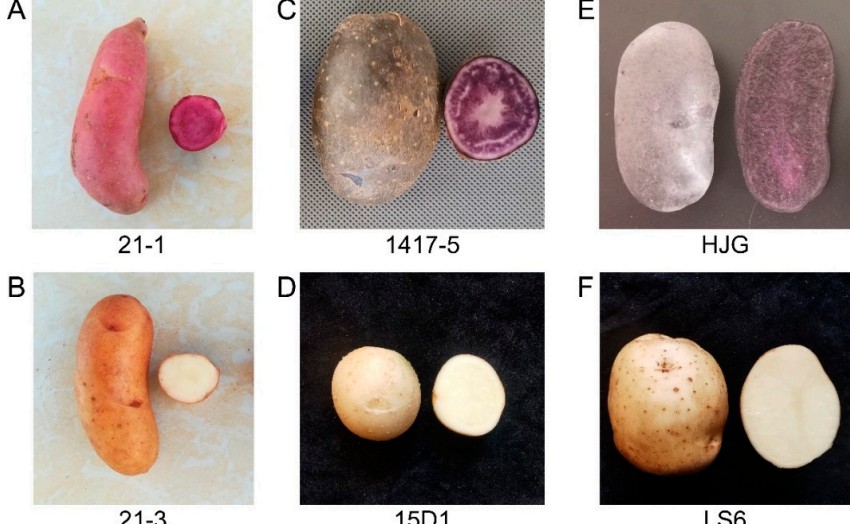

**Figure 1.** The tuber photographs of six potato clones used for reciprocal grafting. (**A**) A red potato clone '21-1' with red skin and red flesh. (**B**) A white potato clone '21-3' with pale yellow skin and white flesh. (**C**) A purple potato clone '1417-5' with purple skin and purple flesh. (**D**) A white potato clone '15D1' with white skin and white flesh. (**E**) A purple potato cultivar 'HJG' with purple skin and purple flesh. (**F**) A white potato cultivar 'LS6' with white skin and white flesh.

## 2.2. Grafting Experiment

In this study, three independent reciprocal grafting combinations, with colored potatoes and white potatoes (21-1 and 21-3, 1417-5 and 15D1, HJG and LS6), were carried out. Virus-free potatoes were planted on the substrate. Healthy seedlings that grew to 7–10 cm with 4–5 leaves were selected for reciprocal grafting. The 'cleft grafting' method was used [27], with 15 grafting seedlings for each combination. After a week of cultivation in a greenhouse with low light, 18–24 °C, and 85–95% humidity, the grafting seedlings were transplanted outdoors. Conventional cultivation processes were performed and tuber samples were collected at maturity (100–110 days after transplanted). The color of the tuber samples was observed and their anthocyanin content was measured.

## 2.3. UV-B Treatment

Virus-free red-fleshed potato 21-1 clones were sown in 30 × 30 cm nutrient bags. When the plants grew to about 15 cm, uniform and healthy plants were selected for enhanced UV-B radiation treatment. Based on our previous study, the average UV-B radiation dose in natural sunlight at the experimental farm is approximately 10 kJ·m$^{-2}$, and the enhanced UV-B radiation dose was 2.5 kJ·m$^{-2}$, with a duration time of 8 h each day (9:00–17:00) [24]. The UV-B lamp type was FLB-5A, customized by Nanjing Huaqiang Electronics Co., Ltd. (Nanjing, China), the power was 40 W, and the wavelength range was 280–320 nm. The UV-B lamp was placed 30 cm above the plants, and the height of the lamp was adjusted with the height of the plants. Photosynthetically active radiation was measured using a UV radiometer (Beijing Normal University Photoelectric Instrument Factory, Beijing, China). Tuber samples were collected after 20, 40, 60, and 80 days of UV-B radiation treatment and natural sunlight, respectively. Representative tubers were collected from five randomly selected individual plants. Immediately after collection, each tuber was peeled with a scalpel, and flesh samples were frozen in liquid nitrogen and stored at –80 °C for subsequent study.

## 2.4. Determination of Total Anthocyanin Content (TAC) and Monomeric Anthocyanidin Content (MAC)

The TAC in the tubers was determined using the pH differential method [23]. Fresh tuber flesh samples weighing 1.0 g were measured and extracted using 30 mL of 1.5 mol/L hydrochloric acid (HCl)/95% ethanol (85:15, *v/v*) solution in an ultrasonic bath (XZ-10DTD Ultrasonic Cleaner, SCIENTZ, Ningbo, China) at 45 °C for 30 min. The supernatant was collected after centrifuging (D3024, DLAB, Beijing, China) at 5000 rpm for 5 min. Then, 1 mL of supernatant was added to buffer solutions with pH 1.0 (0.2 mol/L KCl:HCl = 25:67, *v/v*) and 4.5 (0.2 mol/L NaAc:HAc = 1:1, *v/v*), respectively, and its absorbance was measured at 520 nm and 700 nm using a UV-2401PC spectrophotometer (Shimadzu, Kyoto, Japan). The analytical reagents including HCl, ethanol, KCl, NaAc, and HAc were customized by Tianjin Damao Chemical Reagent Factory. The TAC (mg/100 g fresh weight) was quantified by the content of cyanidin-3-glucoside (Cy3G) in the samples according to the formula: $TAC(mg/100\ g) = \frac{A \times Mw \times DF \times V \times 100}{\varepsilon \times L \times m_f}$, where A was the absorbances' difference value of pH1.0 and pH4.5 at 520 nm and 700 nm; Mw was the molar mass of Cy3G, 449.2 g/mol; DF was the dilution multiple; V was the total volume of the extraction solution, mL; $\varepsilon$ was the average molar extinction coefficient of Cy3G, 26,900 L/(mol·cm); L was the width of the cuvette, 1 cm; and m was the mass of the tuber flesh sample, g.

The MAC in the tubers was analyzed using the HPLC method [28]. A stock solution of six anthocyanidin standards, including cyanidin chloride (≥96%, EXTRASYNTHESE, Genay, France), delphinidin chloride (≥97%, EXTRASYNTHESE, Genay, France), pelargonidin chloride (≥97%, EXTRASYNTHESE, Genay, France), peonidin chloride (≥97%, EXTRASYNTHESE, Genay, France), petunidin chloride (≥95%, EXTRASYNTHESE, Genay, France), and malvidin chloride (≥97%, EXTRASYNTHESE, Genay, France), were prepared using a 10% hydrochloric acid methanol solution, and mixed standard solutions were also prepared. Fresh tuber flesh samples (0.5~1.0 g) were ground to a fine powder in

liquid nitrogen and added to a hydrochloric acid–ethanol extraction solution, subjected to ultrasound extraction for 30 min, hydrolyzed in a boiling water bath (HH-4, JIECHENG EXPERIMENT APPARATUS, Shanghai, China) for 1 h, and then the sample supernatant solution was obtained through a 0.45 μm microporous filter membrane. The mixed standard solution and sample solution were added separately to the liquid chromatograph. A HPLC (1200 s Agilent Technologies, Santa Clara, CA, USA) with a UV detector was utilized under the following conditions for the separation and quantification of anthocyanidins: a total run time of 30 min; a ZORBAX StableBond C18 column (250 mm × 4.6 mm × 5 μm, Agilent Technologies, Santa Clara, CA, USA) maintained at 35 °C; a binary mobile phase consisting of (A) water with 1% formic acid (>99.95% Sigma, St. Louis, MO, USA) (*v/v*) and (B) acetonitrile (>98%, Sigma, St. Louis, MO, USA) with 1% formic acid (*v/v*); gradient elution of 8% B at 0 min, 12% B for 2 min, increasing to 80% B for 15 min, increasing to 20% B for 18 min, decreasing to 8% B for 2 min, and maintained there for 8 min; flow rate of 0.8 mL/min; and injection volume of 20 μL. The absorption peak area of wavelength 530 nm was determined for anthocyanidins. A standard curve of the peak area versus content was prepared via gradient dilution of the mixed standard solution, with a quantification range from 0.5 mg/L to 50 mg/L. The MAC of the sample was determined by considering the retention time and peak area of the chromatogram based on the standard curve. Three biological replicates were performed for each sample. A representative mixed standard solution chromatogram and two tuber flesh sample solutions' chromatograms are shown in Figure S2.

### 2.5. Transcriptome Sequencing and Bioinformatics Analysis

Tuber samples from 20 days of UV-B radiation (UVB) and 20 days of natural light exposure (CK) were selected for transcriptome sequencing. Total RNA was extracted using the Trizol reagent kit (Thermo Fisher, Carlsbad, CA, USA). The RNA libraries were constructed and sequenced on the illumina NovaseqTM 6000 platform by OE Biotech, Inc., Shanghai, China. High-quality clean reads were obtained and aligned to the potato reference genome DM1-3_v6.1 (http://spuddb.uga.edu/index.shtml, accessed on 7 April 2023) using HISAT2 software. The expression value of each potato gene was calculated using normalized FPKM (Fragments Per Kilobase of transcript per Million mapped reads) values (Table S1). Differentially expressed genes (DEGs) were identified using DESeq2 software, with the criteria of |log2foldchange| > 1 and a false discovery rate (FDR) < 0.05. And PCA, volcano plots, heatmaps, as well as a GO (Gene Ontology) enrichment analysis and KEGG (Kyoto Encyclopedia of Genes and Genomes) enrichment analysis of the DEGs were performed using the OECloud platform (https://cloud.oebiotech.com/task/, accessed on 7 April 2023) and TBtools-II platform [29].

### 2.6. Verification of the qRT-PCR (Quantitative Real-Time Polymerase Chain Reaction)

The extracted total RNA was used to synthesize the first strand of cDNA, according to the manufacturer's instructions, using the PrimeScript RT reagent kit and gDNA Eraser (Perfect Real Time) (Takara Biomedical Technology (Beijing) Co., Ltd., Beijing, China). The qRT-PCR was performed using TB Green Premix Ex Taq II (Tli RNaseH Plus) (Takara Biomedical Technology (Beijing) Co., Ltd., Beijing, China). The qRT-PCR-specific primers can be found in Table S2; St*Ef1α* was used as the internal normalization gene. The qRT-PCR procedure was set at 95 °C for 30 s, followed by 40 cycles of 95 °C for 5 s and 60 °C for 30 s. Gene expression levels were evaluated using the $2^{-\Delta\Delta Ct}$ method [30].

### 2.7. Statistical Analysis

In each independent experiment, three biological replicates were performed for both the treatment and the control. Bar charts, line graphs, and pie charts were plotted using GraphPad 8.3.0 software. The independent samples' *t*-test was conducted using SPSS 26.0 software, with $p < 0.05$ considered significant differences. Data were presented as mean ± standard deviation (SD). The correlation coefficient between the qRT-PCR and

RNA-seq was calculated using linear regression methods. Pearson correlation analysis was performed using the 'psych' R package.

## 3. Results

### 3.1. Anthocyanins in Colored Potato Tubers Are Synthesized In Situ

To determine whether tuber anthocyanins were synthesized in the leaves and transported to the tubers, we conducted three reciprocal grafting experiments in groups using colored potatoes and uncolored potatoes. As seen in 21-3/21-1, 15D1/1417-5, and LS6/HJG, the rootstocks are purple or red potato, and the tubers of the grafted plant are purple or red and contain anthocyanins. Inversely, when the rootstock is white potato, the tubers of the grafted plants remain white and without detected anthocyanins, as seen in 21-1/21-3, 1417-5/15D1, and HJG/LS6 (Table 1, Figure S1). These results showed that the color and anthocyanin content of the tubers in grafted plants depend on the rootstock. Anthocyanins in potato tubers are synthesized in situ, not synthesized in the leaves and transported to the tubers, suggesting different biosynthetic pathways for the anthocyanin biosynthesis in potato stems and leaves compared to the tubers.

**Table 1.** Three independent reciprocal grafting combinations and their tuber TAC.

| Potato Clones | Tuber Color (Skin, Flesh) | TAC (mg/100 g FW) | Grafting Plants (Rootstock/Scion) | Tuber Color (Skin, Flesh) | TAC (mg/100 g FW) |
|---|---|---|---|---|---|
| 21-1 | Red, Red | $26.77 \pm 2.34$ | 21-3/21-1 | Red, Red | $35.53 \pm 2.03$ |
| 21-3 | Pale Yellow, White | nd | 21-1/21-3 | Pale Yellow, White | nd |
| 1417-5 | Purple, Purple | $9.71 \pm 0.72$ | 15D1/1417-5 | Purple, Purple | $18.68 \pm 1.83$ |
| 15D1 | White, White | nd | 1417-5/15D1 | White, White | nd |
| HJG | Purple, Purple | $39.79 \pm 0.90$ | LS6/HJG | Purple, Purple | $68.92 \pm 4.04$ |
| LS6 | White, White | nd | HJG/LS6 | White, White | nd |

Note: Three biological replicates were performed for each tuber flesh sample and represented as mean $\pm$ SD. nd denotes not detected.

### 3.2. Enhanced UV-B Radiation in Colored Potato Stems and Leaves Increases the Anthocyanin Content in Tubers

Whether enhanced UV-B radiation in the aerial parts of the potato plant affects the accumulation of anthocyanins in the underground tubers is an interesting research question. Therefore, we conducted an enhanced UV-B radiation experiment on a red-fleshed potato clone '21-1'. The results showed that the red color of the potato's tubers exposed to enhanced UV-B deepened, with a significant increase in its TAC, compared to natural sunlight exposure (Figure 2A,B). With the duration of the radiation increasing, the TAC in the tubers gradually increased, but without a statistical difference in the samples of each after 20 days (Figure 2B). Among them, the highest TAC was found in tubers exposed to 60 days of UV-B radiation: 44.32 mg/100 g FW (Figure 2B). Furthermore, we determined the MAC in the tubers after 80 days of enhanced UV-B radiation via HPLC. Compared to natural sunlight exposure, the UV-B radiation treatment led to a significant increase in both the contents of pelargonidin (Pg) and peonidin (Pn) in the tubers, with the Pg content increasing by 24% and the Pn content by 7.75% (Figure 2D). In particular, the ratio of Pg in the UV-B-treatment tubers increased compared to the untreated control, while the Pn ratio decreased (Figure 2C). These findings demonstrated that enhanced UV-B radiation in the aerial parts of colored potatoes promotes an increase in the TAC of the underground tubers, particularly increasing their Pg content.

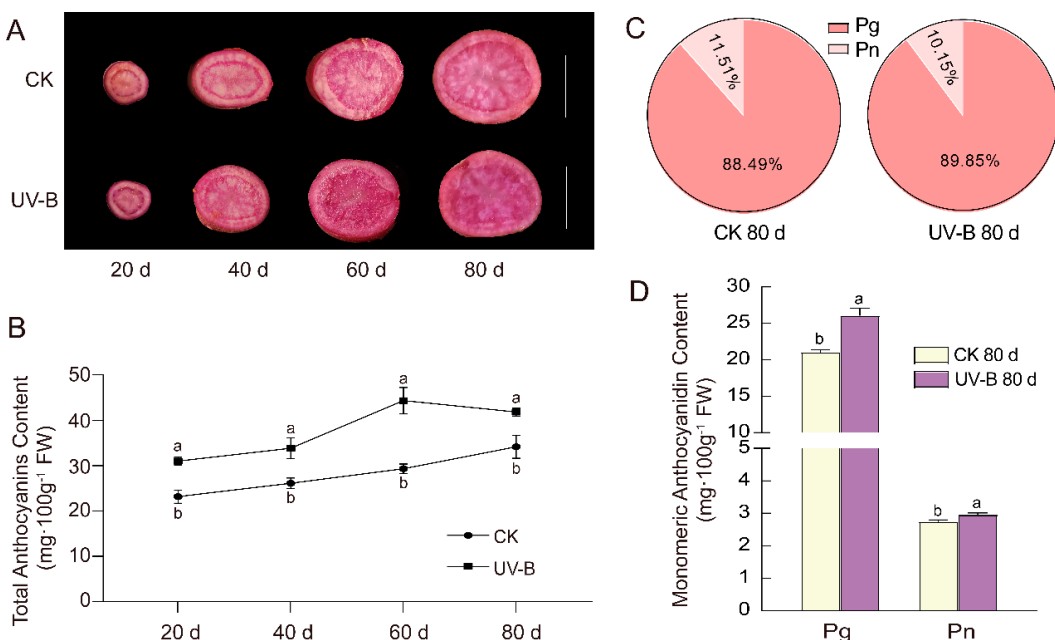

**Figure 2.** The effects of enhanced UV-B radiation on anthocyanin's accumulation in tubers. (**A**) The photographs of red-fleshed potato clones '21-1' after 20, 40, 60, and 80 days of enhanced UV-B and natural sunlight (CK). The line represents 2 cm. (**B**) TAC of red-fleshed potato clones '21-1' after 20, 40, 60, and 80 days of enhanced UV-B and CK. (**C**) The proportion of pelargonidin (Pg) and peonidin (Pn) after 80 days of UV-B treatment and in untreated tubers. (**D**) MAC after 80 days of UV-B treatment and in untreated tubers. The bars in (**B,D**) represent mean ± SD, and the lowercase letters represent significant differences (*t*-test, $n = 3$, $p < 0.05$).

### 3.3. Effects of Enhanced UV-B Exposure of Potato Stems and Leaves on the Gene Expression in Tubers

#### 3.3.1. Transcriptome Overview

To investigate the effects of enhanced UV-B on the gene expression in potato tubers, we identified the gene expression profile of red-fleshed potato '21-1' tubers after 20 days of enhanced UV-B radiation (UVB) and 20 days of natural sunlight (CK) through RNA-seq analysis. We obtained a total of 41.05 Gb clean reads from six samples, with individual sample data ranging between 6.41 and 7.17 G. The Q30 base distribution was between 93.36 and 93.76%, and the average GC content was 43.12%. All clean reads were mapped to the potato reference genome, with a genome mapping rate for each sample ranging from 89.12 to 90.11% (Table S3). The PCA results showed that the three biological repeats were all clustered in the same group, and the three UVB samples were significantly different from the CK samples, with the first two principal components explaining 93.91% and 2.68% of the data variance, respectively (Figure 3A). Altogether, these results indicated the high reliability of the RNA-seq data, making them suitable for subsequent analyses.

We performed a DEGs analysis using DEseq2 software, with FDR < 0.05 and |log2FC| > 1 as the threshold. In total, 2139 DEGs were detected between the UV-B and CK tuber samples, with 1724 and 415 up-regulated and down-regulated genes, respectively (Figure 3B). The expression heatmap of the DEGs also showed that the enhanced UV-B radiation treatment significantly affected gene transcription expression, activating a large number of genes (Figure 3C). Furthermore, we conducted a GO enrichment analysis on the DEGs. The up-regulated DEGs were significantly enriched in 217 GO terms (Table S4), and 20 of the most significant pathways were selected (Figure 4A), including the jasmonic acid (JA) biosynthetic process (GO:0009695), ethylene-activated signaling pathway (GO:0009873), caffeoyl-CoA O-methyltransferase activity (GO:0042409), and flavonoid biosynthetic process (GO:0009813). The down-regulated DEGs were significantly enriched in 82 GO terms (Table S5), and the top three significantly enriched pathways were associated with DNA-binding transcription

factor activity (GO:0003700), heme binding (GO:0020037), and iron ion binding (GO:0009631) (Figure 4B). Like the GO enrichment analysis, we conducted a KEGG enrichment analysis on all DEGs to study the effects of enhanced UV-B radiation on the metabolic pathways in tubers (Tables S6 and S7). The pathway enrichment analysis for up-regulated DEGs indicated that the colored potato tubers under enhanced UV-B were significantly associated with flavonoid biosynthesis (Ko00941), phenylpropanoid biosynthesis (Ko00940), and nitrogen metabolism (Ko00910) (Figure 4C). Both up-regulated and down-regulated DEGs were significantly enriched in the plant hormone signal transduction (Ko04075) and MAPK signaling pathway (Ko04016) (Figure 4C,D). These results suggested that enhanced UV-B radiation potentially regulates anthocyanin accumulation in the underground tubers of colored potatoes through the phenylpropanoid and flavonoid biosynthesis pathways, as well as the plant hormone signal transduction pathway.

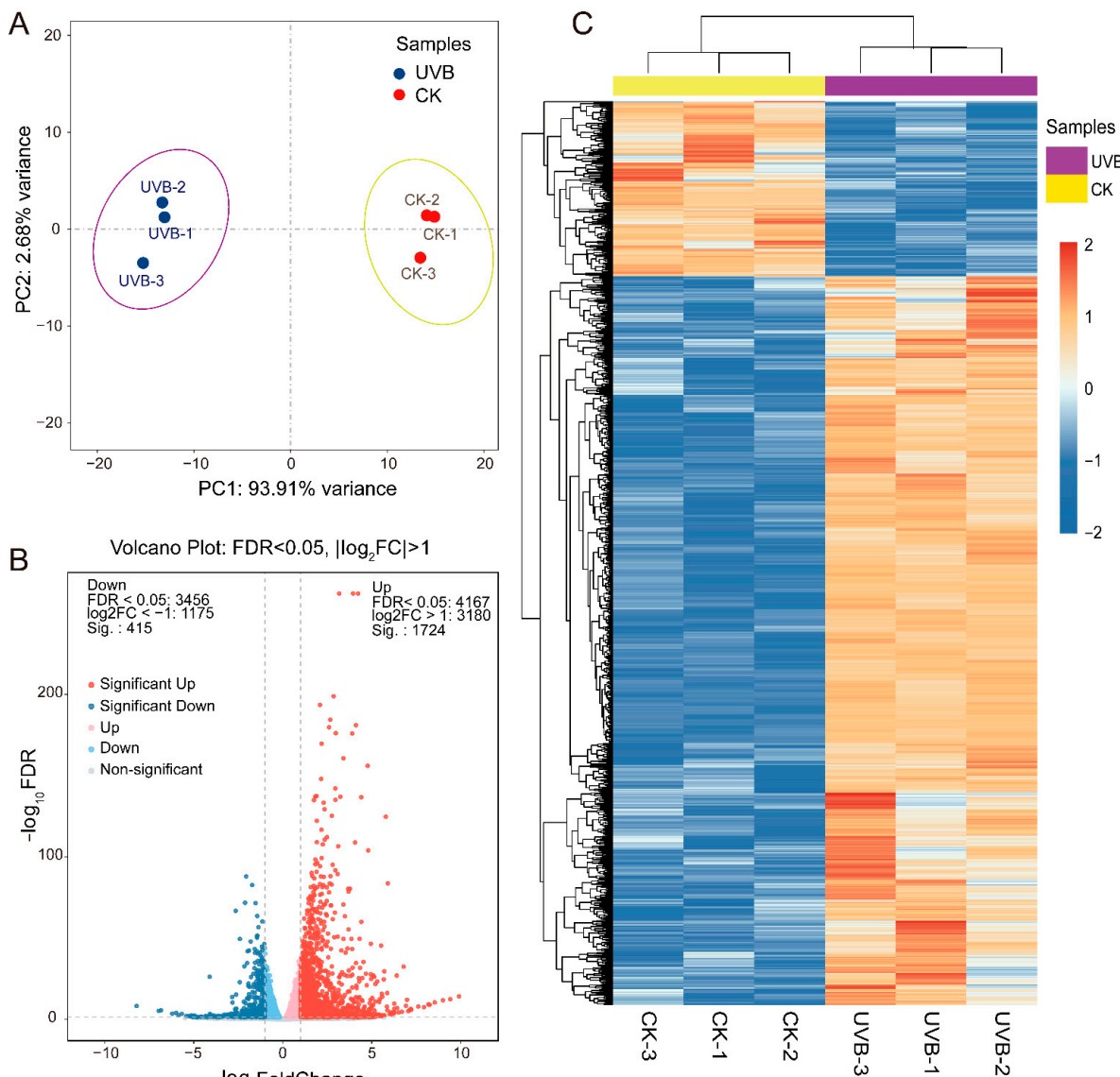

**Figure 3.** Differential expression analysis of genes between UVB and CK tubers via RNA-seq. (**A**) Principal component analysis (PCA) showed differences between UVB and CK tubers. (**B**) Volcano plot illustrated the differentially expressed genes (DEGs) in UVB and CK tubers. (**C**) Heatmap and hierarchical cluster analysis of the DEGs between UVB and CK tubers. The Log$_2$(FPKM + 1) values were row-scaled and displayed according to the color code. The red and blue colors represent the highest (up-regulation) and lowest (down-regulation) expression levels, respectively.

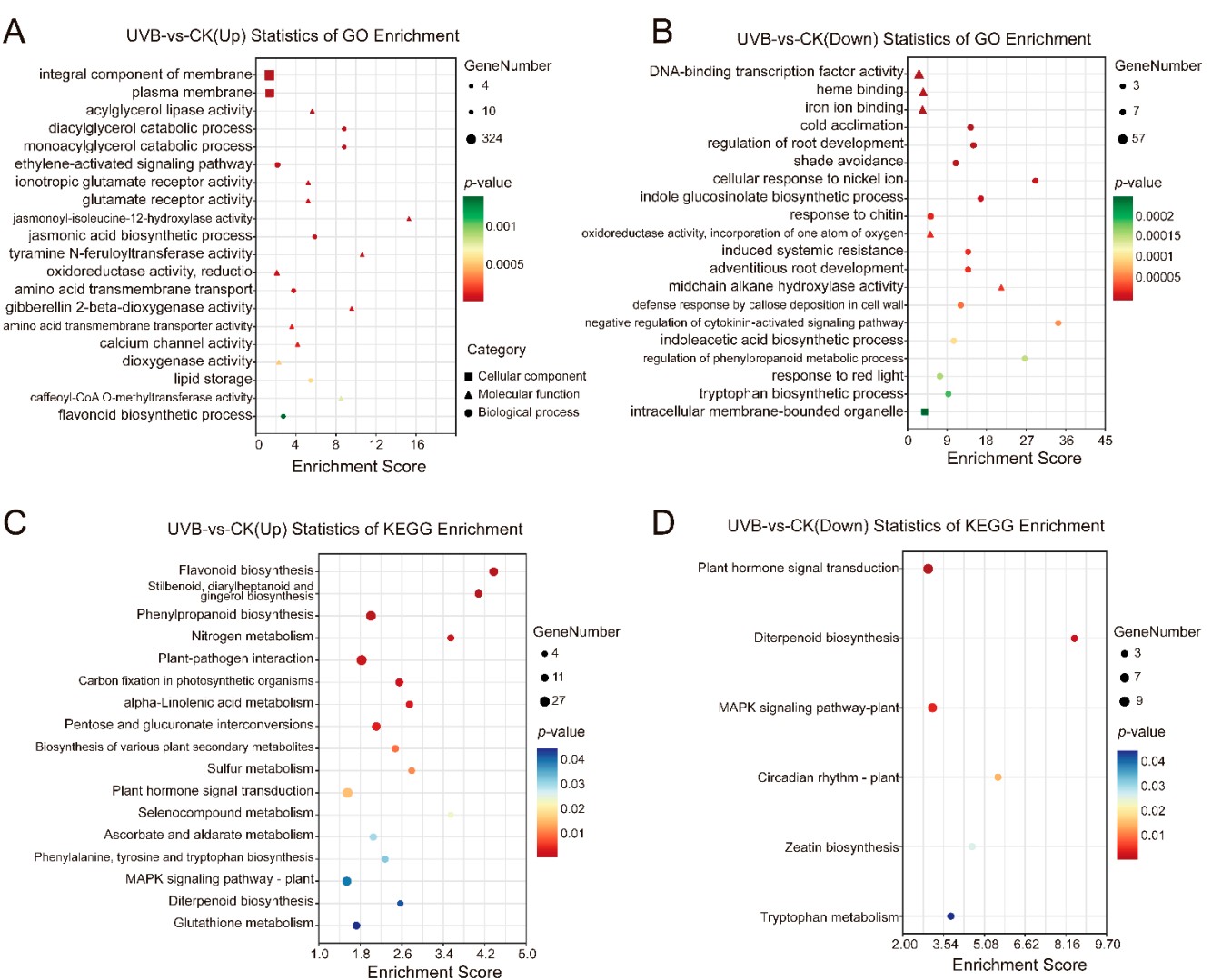

**Figure 4.** Gene ontology (GO) and Kyoto encyclopedia of genes and genomes (KEGG) enrichment analysis of DEGs of UVB and CK tubers. (**A**,**B**) GO analysis of up-regulated and down-regulated DEGs. Top 20 GO terms are displayed for each. (**C**,**D**) KEGG analysis of up-regulated and down-regulated DEGs. All KEGG pathways are shown. The dot size and color indicate the number of genes, and the *p*-value, respectively.

### 3.3.2. Analysis of Anthocyanin Biosynthetic Genes in UVB and CK Tubers

We focused on the expression of anthocyanin biosynthetic enzymatic genes. A total of 30 DEGs in the anthocyanin biosynthesis pathway were screened, including three *PALs*, two *C4Hs*, two *4CLs*, three *CHSs*, one *CHI*, one *F3H*, one *F3′H*, one *F3′5′H*, one *ANS*, five *UFGTs*, three *OMTs*, three *AATs*, three *GSTs*, and two *MATE* gene (Table 2). Compared to CK, UV-B radiation induced a significantly up-regulated expression of all these DEGs in tubers exposed to UV-B, except for *F3′H*, which indicated that the anthocyanin biosynthesis pathway was elevated by UV-B. *F3′H* is responsible for Pn biosynthesis and *F3H* is responsible for Pg biosynthesis based on the flavonoid biosynthesis pathway (Ko00941). The down-regulated *F3′H* and the up-regulated *F3H* could shift the flow of the naringenin substrates towards Pg biosynthesis, which was consistent with the increased Pg content and proportion in tubers exposed to 80 days of UV-B (Figure 2C). Additionally, we also examined the expression levels of the *DFR*, a key rate-limited enzyme gene in the anthocyanin biosynthesis pathway. The expression of *DFR* (*Soltu.DM.02G024900*) in UV-B tubers was higher than in CK tubers, although a significant difference was absent (fold change of 1.803, Table S1). These results indicated that the enhanced UV-B exposure of potato

stems and leaves elevated the gene expressions throughout the anthocyanin biosynthesis pathway, thus promoting the accumulation of anthocyanins in tubers.

**Table 2.** Analysis of DEGs involved in the anthocyanin biosynthetic pathway in UVB and CK tubers.

| Gene ID (Soltu.DM.) | Gene Name | Gene Function Annotation | Fold Change (UVB/CK) | Regulation |
|---|---|---|---|---|
| 03G004920 | *PAL* | | 2.118 | Up |
| 05G026870 | *PAL* | Phenylalanine ammonia-lyase | 5.041 | Up |
| 10G020990 | *PAL* | | 2.957 | Up |
| 05G019180 | *C4H* | *Trans*-cinnamate 4-monooxygenase | 12.765 | Up |
| 06G032860 | *C4H* | | 2.229 | Up |
| 03G032090 | *4CL* | 4-coumarate-CoA ligase | 2.046 | Up |
| 06G024540 | *4CL* | | 2.576 | Up |
| 05G023610 | *CHS* | | 2.526 | Up |
| 09G028560 | *CHS* | Chalcone synthase | 4.110 | Up |
| 09G028570 | *CHS* | | 3.845 | Up |
| 08G011960 | *CHI* | Chalcone isomerase | 3.561 | Up |
| 02G023850 | *F3H* | Flavanone 3-hydroxylase | 2.366 | Up |
| 03G029340 | *F3'H* | Flavonoid 3'-hydroxylase | 0.333 | Down |
| 11G020990 | *F3'5'H* | Flavonoid 3',5'-hydroxylase | 3.169 | Up |
| 08G026700 | *ANS* | Anthocyanidin synthase | 3.437 | Up |
| 01G047230 | *UFGT* | | 7.760 | Up |
| 05G007630 | *UFGT* | | 6.472 | Up |
| 05G007640 | *UFGT* | UDP-glucose:flavonoid 3-O-glucosyltransferase | 6.909 | Up |
| 07G013940 | *UFGT* | | 2.384 | Up |
| 12G002520 | *UFGT* | | 3.848 | Up |
| 03G019830 | *OMT* | | 4.838 | Up |
| 03G019880 | *OMT* | O-methyltransferase | 9.946 | Up |
| 09G025040 | *OMT* | | 2.589 | Up |
| 12G003540 | *AAT* | | 3.158 | Up |
| 12G028580 | *AAT* | Anthocyanin acyltransferase | 14.592 | Up |
| 12G028660 | *AAT* | | 8.012 | Up |
| 01G008340 | *GST* | | 14.654 | Up |
| 02G020850 | *GST* | Glutathione S-transferase | 2.594 | Up |
| 09G001260 | *GST* | | 2.820 | Up |
| 03G018250 | *MATE* | Multidrug and toxic compound extrusion | 2.443 | Up |

### 3.3.3. Identification of Transcription Factors in UVB and CK Tubers

To explore the regulatory roles of transcription factors (TFs) in anthocyanin biosynthetic genes, we identified 175 differentially expressed TF-encoding genes between the UV-B and CK samples, among which the up-regulated and down-regulated genes were 117 and 58, respectively (Figure 5A). These TFs covered various gene families, such as AP2/ERF, MYB, bHLH, C2C2, C2H2, WRKY, MADS, and bZIP. AP2/ERF, MYB, and bHLH had the highest number of TFs, and up-regulated TFs were much more common than down-regulated TFs. The ten *WRKY* and four *LBD* TFs were up-regulated, while the five *bZIP* TFs were down-regulated. Figure 5B depicted the top 20 TFs with the greatest fold changes (FC) between the UV-B and CK comparisons, with only one *bZIP* and one *TCP* being down-regulated while seven *ERF* and four *MYB* TFs were up-regulated.

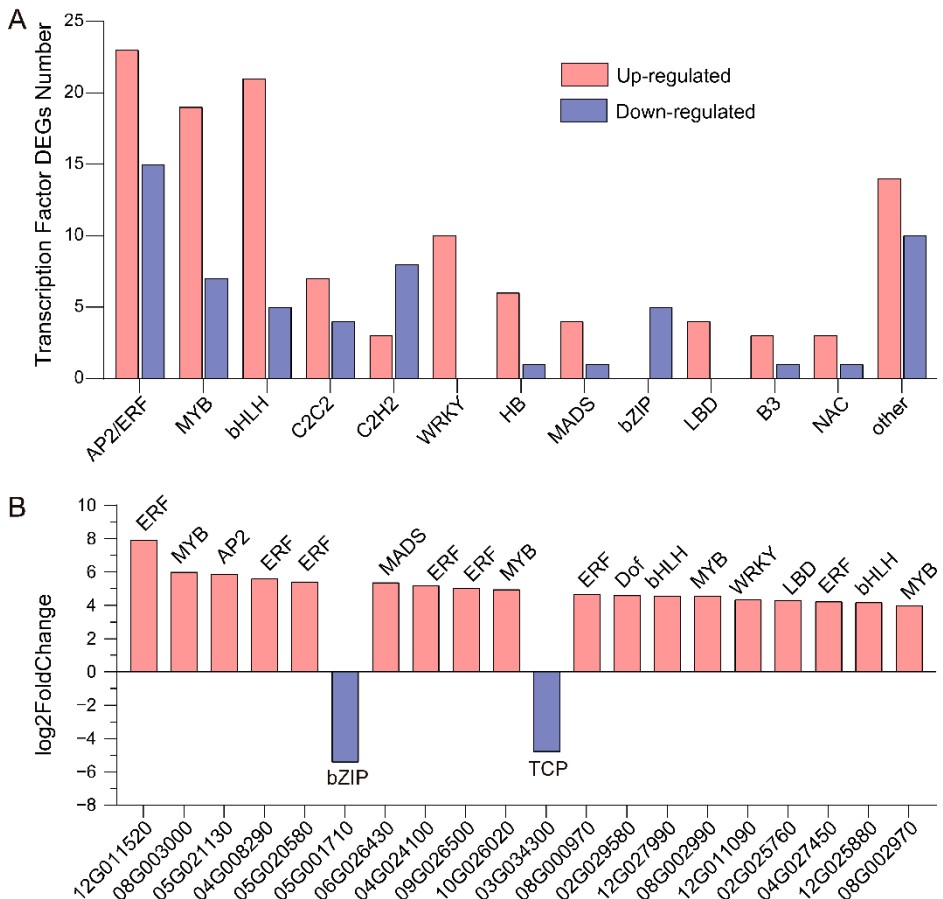

**Figure 5.** Differentially expressed transcription factor (TF)-encoding genes between UVB and CK tubers. (**A**) The number of TF DEGs. (**B**) The top 20 TF-encoding genes with the greatest fold change between UVB and CK tubers. The "Soltu.DM." was abbreviated for every gene name in the x-axis.

MYB TFs have a critical regulatory role in plant anthocyanin biosynthesis. A total of 26 MYB TF DEGs were identified, including 19 R2R3-MYBs and 7 R3-MYBs. The expression heatmap showed that 19 MYB TFs (15 R2R3-MYBs and 4 R3-MYBs) were up-regulated and 7 (4 R2R3-MYBs and 3 R3-MYBs) were down-regulated in UVB tubers (Figure 6A). The correlation analysis of these MYB DEGs with 30 anthocyanin biosynthetic genes revealed that 15 up-regulated MYBs were significantly positively correlated with almost all anthocyanin biosynthetic genes, but negatively correlated with *F3'H*. In contrast, the six down-regulated MYBs were significantly negatively correlated with anthocyanin biosynthetic genes and positively correlated with *F3'H* (Figure 6B). Based on the FPKM expression levels, four R2R3-MYB TF-encoding genes (*Soltu.DM.06G034280_MYB13*, *Soltu.DM.05G005150_MYB105*, *Soltu.DM.06G014570_MYB15*, *Soltu.DM.03G033800_MYB62*) and two R3-MYB TF-encoding genes (*Soltu.DM.06G031130_MYB48_like* and *Soltu.DM.06G004450_MYB48_like*) were potential candidates for the regulation of anthocyanin accumulation in UV-B tubers. In addition to the MYB TFs, *StbHLH1* (*Soltu.DM.09G019660*) also was a transcriptional activator in potato anthocyanin biosynthesis, and it exhibited a significantly up-regulated expression in UV-B tubers (Table S1). These findings suggested that MYB TFs and StbHLH1 play a significant regulatory role in the enhanced UV-B-induced anthocyanin accumulation in colorful potato tubers.

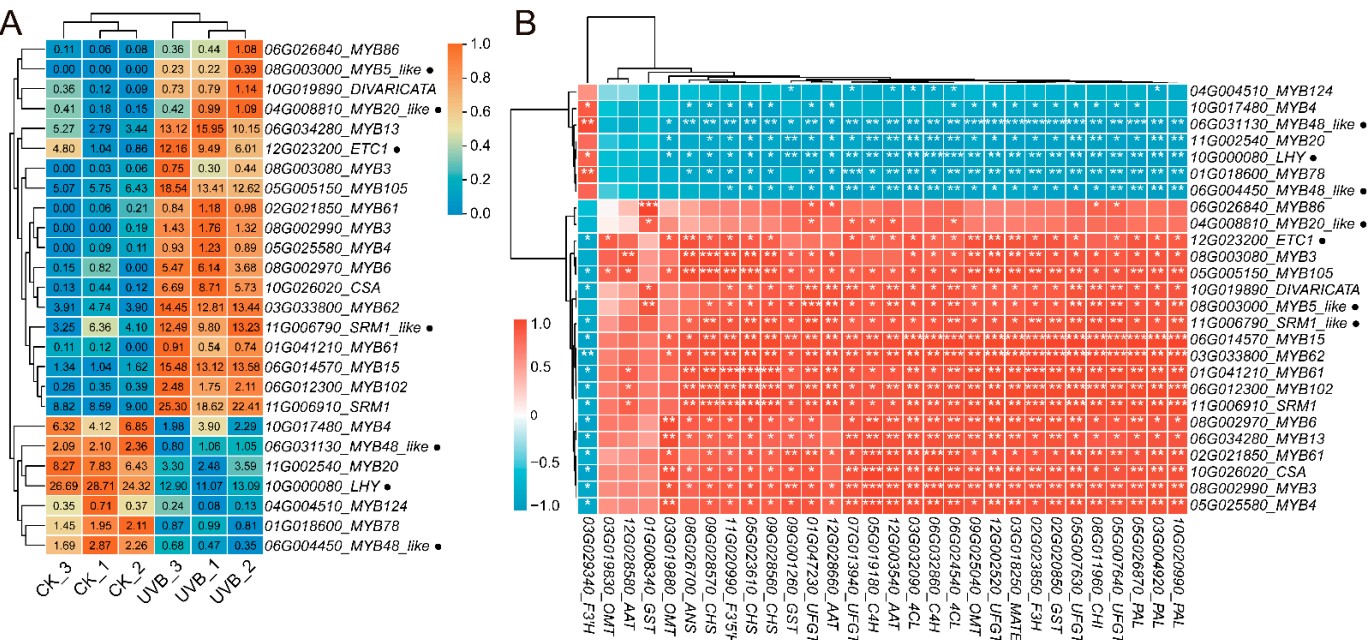

**Figure 6.** Differentially expressed MYB genes between UVB and CK tubers. (**A**) Heatmap and hierarchical cluster showing 26 MYB DEGs. Each column represents a sample, and each row represents a DEG. The numbers in boxes refer to FPKM. The $Log_2$(FPKM + 1) values were row-scaled and displayed according to their color code. The red and blue colors represent the highest and lowest expression levels, respectively. (**B**) Pearson correlation analysis of 26 MYBs and 30 anthocyanin biosynthetic genes. Pearson correlation coefficient values were displayed according to the color code. The red and blue colors represent a positive and negative correlation, respectively. The asterisks in boxes show the *p*-value, * $p < 0.05$, ** $p < 0.01$, and *** $p < 0.001$. Gene names ("Soltu.DM." was abbreviated) with a small black dot represent R3-MYB, the rest belong to R2R3-MYB.

### 3.3.4. Identification of Hormone-Related Genes and UV-B-Related Genes in UVB and CK Tubers

Plant hormones are extensively involved in UV-B stress responses and signal transduction. In the results of the GO enrichment analysis, we identified significant differences between the UVB and CK tubers in the JA biosynthetic process (GO:0009695) and ethylene-activated signaling pathway (GO:0009873) (Figure 4A,B). Thirty-three genes in the ethylene-activated signaling pathway were up-regulated, and ten were down-regulated in UVB tubers (Figure 7A). Thirty AP2/ERF TF DEGs were enriched in the ethylene-activated signaling pathway. In addition, all eight genes involved in the JA biosynthetic process were up-regulated (Figure 7B). The results indicated that ethylene and JA might play a significant role in UV-B tubers responding to enhanced UV-B radiation.

We further investigated the expression levels of genes related to the UV-B stress response, including *UVR8*, *COP1*, *HY5*, and *BBXs* (Figure 7C). A total of 15 *UVR8s*, 2 *COP1s*, 1 *HY5*, and 8 *BBXs* were identified in the potato genome, showing no significant differences in expression between the UVB and CK tubers, except for a significantly down-regulated *Soltu.DM.11G001010_COP1* and a significantly up-regulated *Soltu.DM.04G036140_BBX21*.

### 3.3.5. qRT-PCR Verification

To quantitatively assess the reliability of the expression profiles obtained from the RNA-seq data, we selected 14 genes for qRT-PCR verification. The chosen genes included six DEGs from the anthocyanin biosynthesis pathway, four DEGs from the JA biosynthesis pathway, and four randomly selected DEGs. The results showed that the expression levels of these genes were higher in the UV-B tubers than in the CK tubers (Figure 8A). Most of the quantitative results of the qRT-PCR were consistent with the RNA-seq data (Figure 8B,

$R^2 = 0.7893$), demonstrating that the expression profiles obtained from the RNA-seq data were reliable and that the anthocyanin biosynthesis pathway and JA signal pathway were indeed elevated in the UVB tubers compared to in the CK tubers.

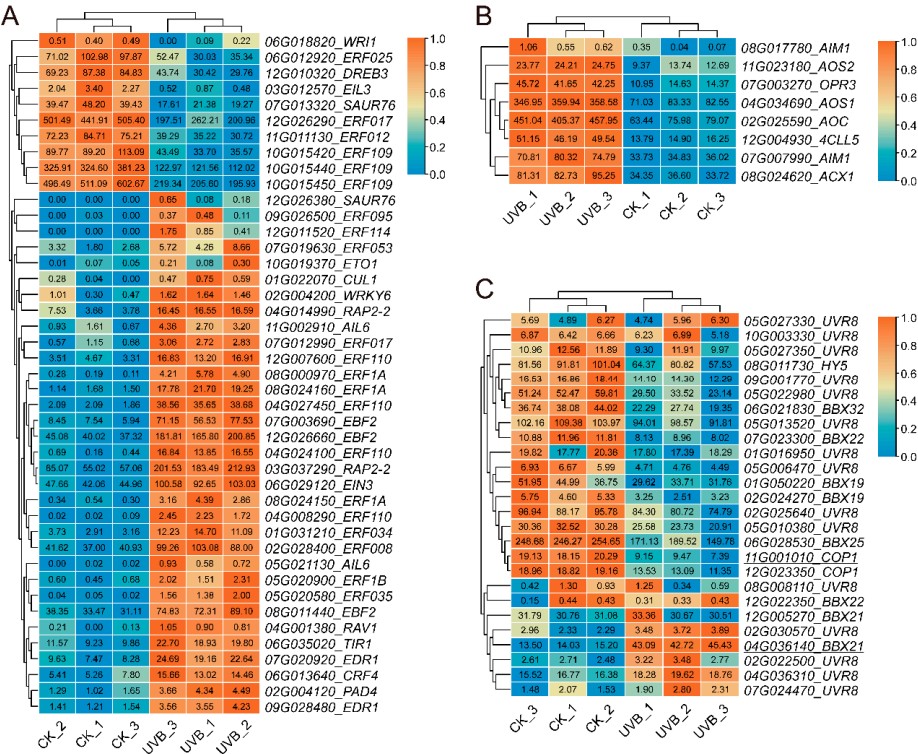

**Figure 7.** Identification of hormone-related and UV-B-related genes between UV-B and CK tubers. (**A**) Heatmap showing 44 DEGs in ethylene-activated signaling pathway (GO:0009873). (**B**) Heatmap showing 8 DEGs in JA biosynthetic process (GO:0009695). (**C**) Heatmap showing 26 UV-B-related genes in potato tubers, including 15 UVR8, 2 COP1, 1 HY5, and 8 BBX. Genes with an underline represent a DEG, the rest display no significant difference.

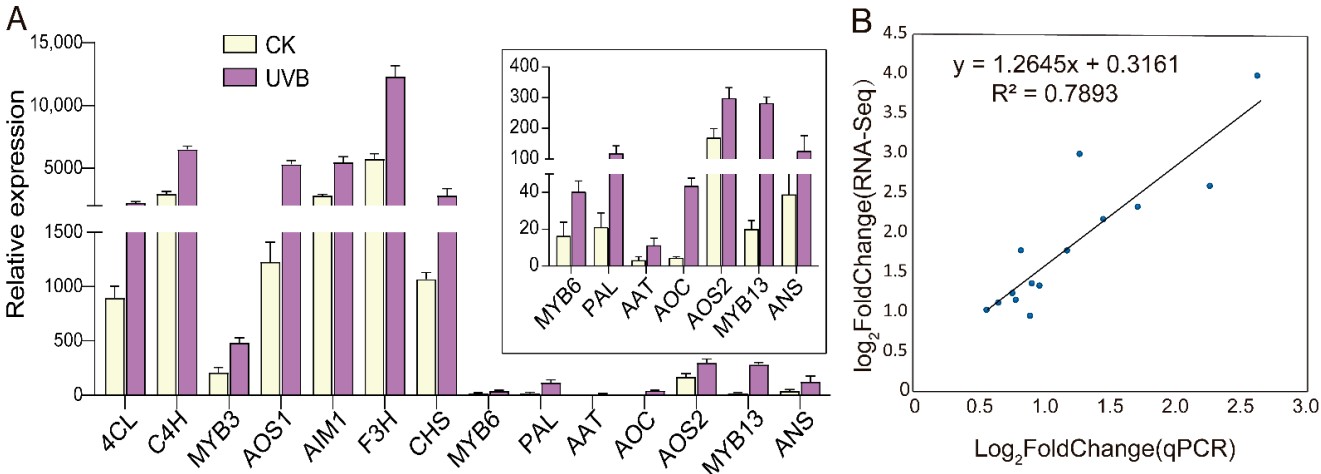

**Figure 8.** Expression analysis of 14 selected genes between UVB and CK tubers. (**A**) Histogram showing the relative expression levels of 14 selected genes. The error bars indicate the standard errors of three biological and three technical replicates. (**B**) Scatter plots show linear regression and $R^2$ between RNA-Seq and qRT-PCR data.

## 4. Discussion

As a crop harvested for its underground tubers, the biosynthesis and accumulation structures of anthocyanin in potatoes are a significant scientific issue. Several studies have hypothesized that the anthocyanins in potatoes were synthesized primarily in the aboveground leaves before being transported and stored in underground tubers [31,32], and that light exposure accelerated the synthesis of anthocyanins in the leaves, as well as their transportation process to the tubers. In this study, we first conducted grafting experiments to ascertain that there was no "synthesis-transport" relationship between the aboveground stem and leaf parts and the underground tuber in terms of anthocyanin accumulation. The trials involving three independent groups of reciprocal grafting with colored and white potatoes demonstrated that the synthesis and accumulation of potato tuber anthocyanin depended on the rootstock, rather than transport from the scion to the rootstock (Table 1, Figure S1). Earlier research also proposed that the rootstock was a critical element in the anthocyanin biosynthesis in potato tubers [33]. These studies confirmed that the anthocyanins in potato tubers are synthesized in situ, negating the existence of a "leaf-tuber" anthocyanin "synthesis-transport" process. A more reasonable explanation for the light-induced promotion of anthocyanin accumulation in potato tubers might be that the light signal molecules regulate anthocyanin biosynthesis via long-distance transport [33,34]. This light signal might also mediate the UV-B radiation affecting potato tuber anthocyanin accumulation. Furthermore, we found that the grafting operation also induces anthocyanin accumulation in the potato tubers (Table 1), which is a discovery worthy of further investigation.

Due to their immobility, plants have evolved complex response mechanisms to adapt to constantly changing environments, such as UV-B radiation reaching the earth's surface due to the ozone hole [35]. Plants mitigated the damage caused by UV-B radiation through the accumulation of secondary metabolites such as flavonoids and anthocyanins [35]. Numerous studies have reported the inducing role of UV-B radiation on anthocyanin synthesis in different tissues of plants, including leaves, petals, and fruits [24,35,36]. A notable characteristic of these studies was the anthocyanins' accumulation in the aboveground parts of the plants (such as leaves, flowers, and fruits) in response to UV-B radiation. However, how the underground organs (as potato tubers), which have indirect exposure to UV-B, respond to the enhanced UV-B radiation in the aboveground parts presents an interesting scientific question. Some studies have initially reported the effects of enhanced UV-B radiation on the physiological and biochemical properties of the underground organs of plants. Müller et al. [22] enhanced UV-B radiation ($21.6 \text{ kJ} \cdot \text{m}^{-2}$) for two weeks in black carrots during their normal growth period, resulting in a significant increase in the total anthocyanin and total phenolic contents in black carrot taproots. Qi et al. [37] observed a significant increase in the tuber anthocyanin content when potatoes growing under normal field sunlight conditions were supplemented with short periods of UV-B radiation ($0.24 \text{ kJ} \cdot \text{m}^{-2}$) daily for 20 days before harvest. Wu et al. [23] found that an increase of $5 \text{ kJ} \cdot \text{m}^{-2}$ of UV-B radiation notably increased the anthocyanin content in potato tubers in artificial climate chambers. Our previous research also showed that the supplementation of $2.5 \text{ kJ} \cdot \text{m}^{-2} \cdot \text{d}^{-1}$ of UV-B radiation to colored potatoes growing under natural sunlight conditions did not significantly affect the growth of the potato plants, but did enhance their antioxidant enzyme activity and the concentration of secondary metabolites in their leaves [24]. In this study, we further investigated the effect of enhanced UV-B ($2.5 \text{ kJ} \cdot \text{m}^{-2} \cdot \text{d}^{-1}$) on the anthocyanin accumulation in potato tubers. The results demonstrated that enhanced UV-B radiation significantly increased the TAC in red-fleshed potato tubers (Figure 1A,B). HPLC analyses revealed that the Pg and Pn contents in tubers after UV-B treatment were significantly higher than those in untreated tubers, with a higher proportion of Pg (Figure 1C,D). Anthocyanins are formed through the glycosidic linkage of anthocyanidins and glycosides, and the six common anthocyanidins include pelargonidin (Pg), cyanidin (Cy), peonidin (Pn), delphinidin (Dp), petunidin (Pt), and malvidin (Mv). Red potatoes mainly contain Pg, while purple potatoes predominantly contain Dp and Pt [38,39]. Hence, the deepened red color of the UV-B tubers

compared to untreated tubers was associated with an increase in the Pg content and its proportion. The findings suggested that enhanced UV-B radiation stimulates the accumulation of anthocyanins not only in aboveground potato leaves but also in belowground tubers. Furthermore, the pathways responsible for anthocyanin synthesis differ between leaves and tubers.

Plants accumulated flavonoid metabolites such as anthocyanins to resist the production of reactive oxygen species (ROS) induced by UV-B [16]. UV-B radiation increased the anthocyanins' content by stimulating the gene expressions of the flavonoid biosynthetic pathway [23,36,40]. As expected, in the transcriptome comparison, the transcriptional levels of the entire anthocyanin biosynthetic pathway genes, including *CHS*, *CHI*, *F3H*, *ANS*, *UFGT*, and *GST* (Table 2), were elevated in UV-B tubers. In particular, *F3'H* expression was down-regulated in UV-B tubers. *F3'H* plays an irreplaceable role in Pn biosynthesis, and its down-regulation directed the naringenin substrates towards the Pg branch, thereby generating an increased proportion of Pg in UV-B tubers. Additionally, anthocyanin biosynthesis is regulated by the MBW transcription factor complex [11,13]. In potatoes, StAN1, StMYBA1, StMYB113, StJAF13, StbHLH1, and StAN11 have been proven to regulate anthocyanin biosynthesis, with StAN1, StMYBA1, and StMYB113 belonging to the R2R3-MYB TFs [7,41–43], StJAF13 and StbHLH1 belonging to the bHLH TFs [7,44], and StAN11 being a WD40 protein [45]. In this study, enhanced UV-B radiation induced the up-regulation of StbHLH1 expression, while the other five TFs did not show significant differences, implying that the StbHLH1 TF was involved in UV-B radiation-induced tuber anthocyanin biosynthesis. Moreover, we found that four R2R3-MYB TFs (*Soltu.DM.06G034280_MYB13*, *Soltu.DM.05G005150_MYB105*, *Soltu.DM.06G014570_MYB15*, *Soltu.DM.03G033800_MYB62*) and two R3-MYB TFs (*Soltu.DM.06G031130_MYB48_like* and *Soltu.DM.06G004450_MYB48_like*) were significantly associated with the anthocyanin biosynthetic genes (Figure 6B), suggesting that these MYB TFs might participate in UV-B radiation-induced tuber anthocyanin biosynthesis [40,46,47]. However, their real functions and regulatory mechanisms require further investigation.

Plant hormones also participated in the plant responses to UV-B [48,49]. UV-B radiation mediated the UV-B stress responses and synthesis of essential metabolites through the JA, salicylic acid (SA), and ethylene (ETH) signaling pathways [47,50,51]. In this study, we identified significant differences in the JA biosynthesis and ETH-activated signaling pathways between UV-B and CK tubers (Figure 3A,B). Enhanced UV-B radiation activated eight genes involved in the JA biosynthetic process and thirty genes in the ETH-activated signaling pathway in UV-B tubers (Figure 6A,B). JA and ETH served as long-distance transport signal molecules to regulate plant growth and development [52,53]. MeJA facilitated the accumulation of anthocyanins under light conditions during the development of radish seedlings [54]. The application of 50 mg·L$^{-1}$ MeJA on red-fleshed potato '21-1' leaves significantly increased the plant's tuber TAC (Figure S3). Therefore, the enhanced UV-B radiation in potato stems and leaves might activate the JA signaling pathway, which consequently promotes tuber anthocyanin accumulation. The typical UV-B-induced signaling pathway involved UV RESISTANCE LOCUS8 (UVR8), CONSTITUTIVELY PHOTOMORPHOGENIC 1 (COP1), ELONGATED HYPOCOTYL 5 (HY5), and B-box (BBX) [18,35,46]. However, no significant differences were observed in the expression of these UV-B-induced genes in the UV-B and CK tubers, except for *Soltu.DM.11G001010_COP1* and *Soltu.DM.04G036140_BBX21*, which were down-regulated and up-regulated, respectively (Figure 7C). The UVR8-COP1-HY5-BBX signaling pathway was involved in translational level regulation [35], which might be the reason why these genes showed no significant transcriptional changes in the tubers. The HY5 protein's function in long-distance transport from the aboveground to the underground parts of plants, regulating growth, development, and resistance, has been extensively researched [55–57]. However, whether this long-distance signaling mediates UV-B-induced tuber anthocyanin biosynthesis at the protein level requires further investigation.

## 5. Conclusions

This study demonstrated, through grafting experiments, that the anthocyanin synthesis in potato tubers is in situ, and that no synthesis–transport process exists from the leaves to the tubers. The enhanced UV-B exposure of potato stems and leaves stimulates the expression of *StbHLH1* and anthocyanin biosynthetic genes in tubers, thus promoting anthocyanin accumulation. Six MYB TFs are significantly correlated with anthocyanin biosynthetic genes. Moreover, the JA signaling pathway might participate in UV-B radiation-induced tuber anthocyanin biosynthesis. These findings lay a foundation for further exploration into the UV-B-induced regulatory network of anthocyanin biosynthesis in potato tubers.

**Supplementary Materials:** The following supporting information can be downloaded at https://www.mdpi.com/article/10.3390/cimb45120621/s1, Figure S1. The photographs of the reciprocal grafting combinations with colored and uncolored potatoes; Figure S2. A representative mixed standard solution chromatogram and two tuber flesh sample solutions' chromatograms; Figure S3. Exogenous methyl jasmonate (MeJA) spraying treatment; Table S1. All genes' FPKM from RNA-seq; Table S2. The list of primers used for qRT-PCR; Table S3. Statistical summary of RNA-seq and its mapping rate; Table S4. Statistics of GO enrichment of 1724 up-regulated DEGs; Table S5. Statistics of GO enrichment of 415 down-regulated DEGs; Table S6. Statistics of KEGG enrichment of 1724 up-regulated DEGs; Table S7. Statistics of KEGG enrichment of 415 down-regulated DEGs.

**Author Contributions:** H.G. conceived the idea and designed the research. L.C. performed the experiments. L.C. and M.L. wrote the manuscript. M.L. and X.Z. contributed to visualizing the data. Z.G., K.L. and Y.S. contributed to data collection. H.G. and Q.W. contributed to reviewing the manuscript and approved the final manuscript. All authors have read and agreed to the published version of the manuscript.

**Funding:** This work was supported by the National Natural Science Foundation of China (31860402) and the Science and Technology Projects in Yunnan Province (202102AE090018).

**Institutional Review Board Statement:** Not applicable.

**Informed Consent Statement:** Not applicable.

**Data Availability Statement:** The data are included within the article, and the RNA-Seq datasets presented in this study can be found in the National Center for Biotechnology Information (NCBI) sequence-read archive with accession number PRJNA1028809 (https://www.ncbi.nlm.nih.gov/bioproject/PRJNA1028809/).

**Conflicts of Interest:** The authors declare no conflict of interest.

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
