# Peer review of "Enhanced UV-B Radiation in Potato Stems and Leaves Promotes the Accumulation of Anthocyanins in Tubers"

_cimb, doi:10.3390/cimb45120621_

Round 1

Reviewer 1 Report

Comments and Suggestions for Authors

The manuscript “Enhanced UV-B Radiation of Potato Stems and Leaves Promotes Anthocyanins Accumulation in Tubers” fits the journal’s scope. In the present manuscript, the authors present their results on the influence of the effect of UV-B radiation on the biosynthesis of anthocyanins at potato tuber and investigated the potential mechanisms implicated. The research design is clearly stated. The materials and methods are described in sufficient detail, and cited accordingly. The quality of presentation of the results is meets the criteria of the journal. The results are discussed in relation to other findings, and the novelty of the research, as stated in the introduction, is emphasized in the discussion section. The conclusions of the present research are sustained by the results presented.

The present manuscript is suitable for publication after minor (but mandatory!) corrections.

Section 2.4 – please describe in detail the HPLC method.

Section 2.4 – please present in separate paragraphs the UV-Vis and HPLC methods. Sample preparation, equipment, standards, conditions, number of samples should be addressed,

Please insert tables and figures in text.

Table 1 – please indicate the number of replicates; please indicate that average values are presented with SD; please indicate for Pale yellow, White if the results were conducted. Is 0 the value obtained?

Lines 44-46 – please rephrase

Line 83 – please correct the error

Reviewer 2 Report

Comments and Suggestions for Authors

The manuscript delves into the repercussions of heightened UV-B radiation on anthocyanins biosynthesis in potato tubers. Through reciprocal grafting experiments and UV-B radiation treatments on red-fleshed potato plants, the study explores the in situ synthesis of anthocyanins in tubers, distinct from transport from leaves. The outcomes spotlight a significant increase in the total anthocyanins content, monomeric pelargonidin, and peonidin in the studied potato tubers under enhanced UV-B radiation. Transcriptomic analysis identified 2139 differentially expressed genes. The discussion section targeted the experimental findings, providing insights into in situ anthocyanins synthesis and the intricate regulatory network prompted by heightened UV-B radiation. To fortify the conclusion, a more explicit summary of key findings and their implications, coupled with a delineation of contributions and potential avenues for future research, would augment its impact.

However, there are some issues to address for an improved version, hence please consider the following suggestions:

L.30 – delete JA – not used later in „Abstract”;

L.38 – use italics for Latin names;

In #2.3, clarify the UV-B treatment details, including devices, producers, and experimental details, including the used device(s) + producers and experimental arrangement;

L.153 – specify the composition of buffers;

L.154 – provide the type & producer for the spectrophotometer;

L.157 – 159 – improper reference – rephrase;

#2.4 – add relevant details about HPLC analysis (equipment & producer, column, mobile phase, separation conditions, calibration range); add a representative chromatogram to prove the separation of stated standards and at least two chromatograms to illustrate the separations accomplished for real samples. Besides, explain why you used six standards (you only reported two compounds from these).

#2 – add a paragraph to describe all the materials used in this research (type & provider);

L.165 – 166 – improper technical language – rephrase;

L.219 – delete „total anthocyanins content” as it is defined in L.149;

L.224 – delete „monomeric anthocyanidins content” as it is defined in L.150;

L.254 – provide the acronym JA for jasmonic acid and use it consistently from that point onward – L.323, 327, 328;

#4- an important part of text within this section has to be re-located to „Introduction” to maintain focus on findings; keep only those issues which are related to your findings here respecting the sequence your finding < > previous findings;

L.362 – rephrase the subjective position by replacing "Interestingly"

L.390 – replace „total anthocyanins content” > TAC as it is defined in L.149;

L.427 – delete „”jasmonic acid”;

L.438 – 439 – evaluate the necessity of the "all caps" format;

L.461, 474 – delete „total anthocyanins content” as it is defined in L.149.

Reviewer 3 Report

Comments and Suggestions for Authors

The authors describe their work on pigments in potatoes. It is interesting and there has been a lot of good work already performed, but I have a few points that should be addressed prior to publication:

In lines 531-532 the authors state: "Data Availability Statement:The data in this paper are available from the corresponding author upon reasonable request.: - This is totally unacceptable in the current era of research. If there is data relevant to the work, it should be hosted in a public repository that anyone can access and get a permanent DOI.

Here are some comments on the writing:

Introduction

  • The introduction lacks a clear outline of the specific gaps in knowledge being addressed. The scope should be narrowed to focus on 1-2 key questions to be investigated.
  • More background is needed on prior related research, especially studies on UV-B effects on underground plant parts like tubers. This provides context.
  • The rationale for studying colored potatoes should be explained. What are the broader impacts?

Methods

  • More details are needed on the UV-B setup, radiation dose, treatment duration, controls used, growth conditions etc. This is essential for reproducibility.
  • The statistical analysis approaches should be described for data like anthocyanin content, gene expression etc.
  • More information is needed on data processing and analysis methods for the transcriptomics experiment. How were differentially expressed genes identified?

Results

  • The results contain a lot of text that could be better presented using figures, tables, charts. This would improve readability.
  • It is unclear which findings are novel contributions vs confirming previous work. This distinction should be made clear.
  • The sequence of results should follow a logical flow. For example, the grafting results seem important but are presented late.

Discussion

  • The discussion should focus on interpreting the key new findings, not just repeating the results. Comparisons to relevant literature should be included.
  • Limitations of the study design should be addressed along with future work. This demonstrates critical analysis.

Overall the manuscript in its current form lacks focus, and many standard components of scientific writing like articulating the specific research gaps, statistical methods, data analysis details, are missing or unclear. Significant revising with a tighter scope and improved clarity in reporting methods and results would strengthen the work.

Comments on the Quality of English Language

A thorough copy-editing should be done

Round 2

Reviewer 2 Report

Comments and Suggestions for Authors

In spite of improvements, this version of the manuscript still exhibits several drawbacks requiring attention, some of which persist from the previous review:

L.82: elaborate on the meaning of 'Proper does UV-B radiation.'

#2 - add a paragraph detailing all materials utilized in this research, specifying their types and providers (e.g., HCl, ethanol, KCl, etc.).

L.164: reframe the technical language concerning 'followed by ultrasonic' for better clarity.

#2.4: include the types and manufacturers of the centrifuge, ultrasonic bath, and water bath; additionally, use an equation editor, like Microsoft Equation Editor, for presenting decent formulas.

L.198–199: improper technical language “The MAC of the sample was determined by considering the retention time and peak area.” - rephrase

Comments on the Quality of English Language

Only minor editing of English language are required

Reviewer 3 Report

Comments and Suggestions for Authors

The authors failed to address my initial comments and appear to have put little to no thought in improving the manuscript in the revision. The English is so poor I cannot even understanding the reasoning. They have even neglected to answer my points, instead they wrote their own version of my points. Totally unacceptable. 

Most importantly: where is the data!? If they did transcriptomics and metabolomics experiments, surely there must be many huge datasets to release? This is completely unacceptable. This data needs to be in public repositories and made available to the reviewers. 

Comments on the Quality of English Language

English is somehow worse than the original submission. 

Round 3

Reviewer 3 Report

Comments and Suggestions for Authors

The authors have addressed my comments, I have no further comments

Comments on the Quality of English Language

Make sure to double check all of the English going forward